# Substitution of Chemical Fertilizer with Organic Fertilizer Affects Soil Total Nitrogen and Its Fractions in Northern China

**DOI:** 10.3390/ijerph182312848

**Published:** 2021-12-06

**Authors:** Md Elias Hossain, Xurong Mei, Wenying Zhang, Wenyi Dong, Zhenxing Yan, Xiu Liu, Saxena Rachit, Subramaniam Gopalakrishnan, Enke Liu

**Affiliations:** 1Institute of Environment and Sustainable Development in Agriculture, Chinese Academy of Agricultural Sciences, Beijing 100081, China; elias.abot@gmail.com (M.E.H.); dongwenyi@caas.cn (W.D.); yzxwell@163.com (Z.Y.); liuxiu@caas.cn (X.L.); liuenke@caas.cn (E.L.); 2Key Laboratory of Dryland Agriculture, Ministry of Agriculture and Rural Affairs of the People’s Republic of China (MARA), Beijing 100081, China; 3Department of Agricultural Botany, Faculty of Agriculture, Sher-e-Bangla Agricultural University, Dhaka 1207, Bangladesh; 4Key Laboratory of Agricultural Environment, Ministry of Agriculture and Rural Affairs of the People’s Republic of China (MARA), Beijing 100081, China; 5Institute of Dryland Farming, Hebei Academy of Agriculture and Forestry Sciences, Hengshui 053000, China; zxm.0223@163.com; 6International Crops Research Institute for the Semi-Arid Tropics (ICRISAT), Patancheru 502324, India; r.saxena@cgiar.org (S.R.); s.gopalakrishnan@cgiar.org (S.G.)

**Keywords:** organic manure, nitrogen fertilizer, soil total N, labile organic N, mineral N, soil fertility

## Abstract

The impact of chemical to organic fertilizer substitution on soil labile organic and stabilized N pools under intensive farming systems is unclear. Therefore, we analyzed the distribution of soil total N (STN), particulate organic N (PON), microbial biomass N (MBN), dissolved organic N (DON), and mineral N (NO_3_^−^ and NH_4_^+^) levels down to 100 cm profile under wheat–maize rotation system in northern China. The experiment was established with four 270 kg ha^−1^ N equivalent fertilizer treatments: Organic manure (OM); Organic manure with nitrogen fertilizer (OM + NF); Nitrogen fertilizer (NF); and Control (CK). Results found that the OM and OM + NF treatments had significantly higher STN, PON, MBN, DON, and NO_3_^−^ contents in 0–20 cm topsoil depths. Conversely, the NF treatment resulted in the highest (*p* < 0.01) DON and NO_3_^−^ depositions in 40–100 cm subsoil depths. The NH_4_^+^ contents in selected profile depths were significantly highest (*p* < 0.01) under OM treatment. The correlations between STN and its fractions were positively significant at 0–10 and 10–20 cm topsoil depths. Our results suggest that partial substitution of chemical fertilizer with organic manure could be a sustainable option for soil N management of intensive farming systems.

## 1. Introduction

Soil nitrogen (N) availability influences the yield, grain N recovery and protein content of cereal crops [1,2,3]. Therefore, N fertilization is inevitable for maintaining crop productivity and grain nutritional quality of cereal-based dryland farming systems where N is often yield-limiting [4,5]. However, in wheat–maize growing dryland farming areas of North China Plain (NCP), the application of chemical N fertilizer (often overdosed) has become a regular practice for decades to achieve higher yields [6]. Such long-term fertilization with chemical fertilizer has aggravated acidification, nutrient imbalance, enzyme activities, and compaction of soils, thereby suppressing crop growth [7,8,9,10]. Moreover, due to low nitrogen use efficiency (NUE; ~33% on average) of major cereal crops most of the unutilized N often leaks out from the farming systems in various N forms, contributing to environmental contamination, global warming, and human health issues [11,12,13,14]. On the other hand, repeated application of organic fertilizer (i.e., manures) was reported to be beneficial for soil organic matter (SOM) content, pH buffering, aggregate formation, water holding and nutrient retention capacity. Moreover, crop productivity is identical when equal total nutrients are supplied from organic or chemical sources [15,16]. Therefore, organic manure substitutions for chemical fertilizer have been suggested as a viable approach to ensure sustainable future food security, to restore soil fertility and structural properties, and to reduce environmental impacts of chemical fertilizer [17,18,19,20]. Still, their impacts on soil labile organic and stabilized N pools under intensively managed dryland farming systems is less understood.

Although nitrogen is a highly mobile element in soil, soil total N (STN) status changes relatively slowly due to its large pool size. Therefore, STN changes are most frequently reported under long-term soil fertility studies [21,22]. Although continuous N input increases STN content in the profile, in topsoil depths, significantly higher total N is often reported with organic or organic–inorganic combined fertilization [17,22,23]. The reason for that can be attributed to organic fertilizer induced increases in soil organic matter (SOM) and available soil N content, because STN status responds to SOM content positively and correlates strongly with available soil N content [24,25]. The labile organic fractions of STN are actively involved in soil N mineralization and considered as sustainable soil fertility indices, which are described as particulate organic N (PON), microbial biomass N (MBN), and dissolved organic N (DON). The dynamics of these pools vary temporarily, respond sensitively to soil fertility management approaches, and affect the short and long-term N supply [25,26,27].

Despite variations, soil mineral N pool (NO_3_^−^ and NH_4_^+^) represents only a small fraction of STN which are readily available for plant acquisition. Application of chemical N fertilizer increases mineral N contents in the profile, but it often exhibits a potential risk of NO_3_^−^ leaching and groundwater contamination [28]. In fact, significant NO_3_^−^ leaching has been reported recently under chemically managed wheat–maize rotations from this area [29]. In contrast, manure-containing treatments have been found to increase the mineral N content in topsoil layers while reducing the NO_3_^−^ leaching by influencing N immobilization processes [28,29,30,31]. Therefore, chemical fertilizer substitution with organic fertilizer could be a sustainable option to reduce N losses.

Soil labile organic N pools are more sensitive than STN to agricultural management practices [32]. These pools are actively involved in short-term N transformations and play a significant role in soil mineral N supply [32,33]. Despite their importance in soil nutrient pools, soil fertility studies often focused on inorganic N pools (NO_3_^−^ and NH_4_^+^) [29,34]. In recent years, however, DON has been getting special attention among scientists due to its substantial contribution to N leaching pathways of forest, grassland, as well as agroecosystems [35,36,37,38,39,40]. For example, in Europe, average DON leaching from agricultural ecosystems accounted for about 26% of total dissolved N leached [41]. In another study in Australia, DON makes up to 40% of the deep drainage nitrogen from irrigated Vertosol cotton-wheat–maize production systems [42]. More recently, significant DON leaching has been reported at high N rates from maize–legume cover rotations under Mediterranean course soil conditions [40]. However, the MBN and PON contents of cultivated soils have been identified greater under continuous manuring or organic–inorganic balanced fertilization than exclusively applied fertilizer N [43]. Nevertheless, only a few fertility studies investigated the distribution of labile organic N pools in cultivated soil in diverse environments, some of which only examined topsoil depths. Therefore, information is limited to conclude the effects of chemical fertilizer substitution with organic manure on the distribution of labile organic N pools in the whole soil profile under intensively managed farming systems.

The crop productivity and sustainability of the agroecosystems are highly dependent on the short-term (seasonal) and long-term (years to decades) dynamics of SOM, including the turnover of soil labile organic fractions and the regeneration of stabilized nutrient pools [44]. Understanding the dynamics of these pools is essential for long-term soil fertility management decisions. Although, recent soil management studies reported that continuous substitution (partial or total) of chemical fertilizer with organic fertilizer significantly increases SOM status, SOC and STN content [22,43,45,46]. Still, their impact on the size and distribution of soil N pools especially those labile organic N pools in the whole soil profile under intensively managed wheat–maize farming systems of northern China is unclear. Despite being an indicator of overall soil quality, SOM can be insensitive to new management practices, and STN changes in the soil profile could be prolonged [47]. Therefore, in this study, we aimed: (1) to investigate the effects of organic manure substitutions for chemical fertilizer on size and distribution of STN and its labile organic (PON, MBN, and DON) and mineral (NO_3_^−^ and NH_4_^+^) fractions in the whole soil profile; and (2) to evaluate how such fertilizer substitution treatments affect the correlations among STN and its fractions; under intensively managed farming system.

## 2. Experimental Methods

### 2.1. Experimental Site

The field experiment was conducted at the dryland water-saving experimental station of Institute of Dryland Farming, Hebei Academy of Agriculture and Forestry Sciences, Hengshui, Hebei province of China (115°10′~116°34′ E and 37°03′~38°23′ N). The site is located at North China flat plain between 17.5 and 28 m above sea level and belongs to the warm temperate semi-humid climate with a continental monsoon. The recorded annual mean precipitation was 497.0 mm, unevenly distributed, mostly fallen in summer months (July to September), while winter months received 120–160 mm. The annual mean temperature, sunshine duration, evapotranspiration, and frost-free period were 12.8 °C, 2509.4 h, 1785.4 mm, and 201 days, respectively. The soil is deep, slightly alkaline (pH 7.8), loamy, and classified as ‘Fluvo-aquic’ according to the FAO-UNESCO system of soil classification [48].

At the beginning of this experiment, the initial soil samples were collected from 0–20 cm topsoil profile, and soil organic matter, alkali-hydrolysable N, available P, available K, and bulk density were recorded as 1.65%, 71.90 mg N kg^−1^, 22.6 mg P kg^−1^, 171.6 mg K kg^−1^, and 1.48 g cm^−3^, respectively.

### 2.2. Experimental Design 

Four N equivalent treatments were established in 2014 based on farmers’ recommended rate of N (270 kg ha^−1^) for dryland farming systems of northern China [49]. The treatments were as follows: (1) Organic manure (OM), 100% N from composted cattle manure; (2) Organic manure with nitrogen fertilizer (OM + NF), 50% N from composted cattle manure plus 50% N from urea; (3) Nitrogen fertilizer (NF), 100% N from urea; and (4) Control (CK), with zero N fertilization. Three replicates for each treatment were arranged in a 17.4 × 10 m^2^ plot maintaining 0.8 m spacing between plots and borders.

The irrigated winter wheat (*Triticum aestivum* L.) and rain-fed summer maize (*Zea mays* L.) rotation system was followed with alternation of wheat varieties (HengH1401, water-saving; and Cangmai6005, drought-resistant) in each calendar year. Crops were harvested each year, and following each harvest, the maize crop residues were removed while the wheat straws were returned to the field for recycling. After each harvesting of summer maize between early to mid-September, a 15–20 cm deep moldboard ploughing was practiced. Manure treatments which contained 341.6 g organic C kg^−1^, 19.1 g N kg^−1^, 10.1 g P kg^−1^, and 8.0 g K kg^−1^, were applied before ploughing and during the soil preparation for winter wheat. The N equivalence of organic manure to urea fertilizer (N 46%) was assessed before each application based on the total Kjeldahl nitrogen content of composted cattle manure. For OM + NF and NF treatments, 40% of total urea fertilizer was applied before sowing as a basal dose and the remaining 60% was side dressed during returning green stage of wheat development. Besides, 57.6 kg P ha^−1^ from triple superphosphate (P_2_O_5_ 46%), and 68.5 kg K ha^−1^ from potassium sulfate (K_2_O 60%) were also applied to wheat across all treatments. Each year, winter wheat was sown between 1 and 10 October at 330 seeds/m^2^ with 15 cm row spacing using a mechanical seeder and harvested at maturity in early June. Small-scale sprinklers were used for wheat irrigation at, before sowing (120 mm), returning green stage (80 mm), and heading stage (80 mm). After winter wheat harvest, summer maize (Zhengdan958) was planted by a mechanical planter with minimum tillage following a 90 mm supplemental irrigation. Chemical pest control measures were applied to control weeds and insect pests.

### 2.3. Soil Sampling 

Soil samples were collected on 27 September 2019 following the harvest of summer maize. Soil cores down to 100 cm soil profile from three random points in each treatment plot were sampled using a standard auger (8 cm diameter). Each soil core of the selected profiles was separated into successional sub-samples at 10 cm depth interval. The three sub-samples of each depth category from each treatment plot were thoroughly mixed to make a composite sample. After removing organic stubbles, each composite sample was divided into two parts. One part was air-dried and kept for analysis of soil chemical properties, and the second part of the fresh sample was passed through a 2 mm sieve and stored at 4 °C for biochemical analysis.

### 2.4. Soil Analysis 

Air-dried soil was used for the analysis of STN, SOC, and PON. For analysis of STN and SOC, soil samples were grounded and passed through a 0.15 mm sieve. STN was determined following the Kjeldahl digestion-distillation procedure as recommended by Bremner and Mulvaney (1982) [50]. SOC was estimated using the K_2_Cr_2_O_7_ oxidation-titration method as described by Blake (1965) [51]. Briefly, 0.1 g sieved soil was digested by boiling with 5 mL 0.8 M K_2_Cr_2_O_7_ and 5 mL concentrated H_2_SO_4_ in an oil bath at 180 °C for 5 min. The organic C content in the digested soil-solution was measured by titration method using 0.2 M FeSO_4_·7H_2_O in presence of C_12_H_8_N_2_·H_2_O indicator. PON was measured using the procedure as described by Bronson et al. (2004) [52]. Briefly, 20 g of air-dried soil (<2 mm) was dispersed in 60 mL sodium hexametaphosphate solution (5 g L^−1^) using a reciprocal shaker for 16 h. The soil suspension was rinsed with deionized water and passed through a 0.15 mm (53 μm mess) sieve. The remnant materials on the sieve were oven-dried and weighed after removing visible stones. The total nitrogen content in PON was estimated following the same procedure for STN determination.

The MBN, DON, and mineral N (NO_3_^−^ and NH_4_^+^) content in soil was measured from fresh soils. MBN was determined using the fumigation-extraction method of Vance et al. (1987) [53]. In brief, two sets of 15 g soil samples were re-wetted and incubated for 24 h in the dark at 25 °C and 40–45% water holding capacity. One set of soil samples was subjected to CHCl_3_ fumigation, and the next one was CHCl_3_-free. After incubation, organic nitrogen in soil samples was extracted with 45 mL 0.5 M K_2_SO_4_ solution, shaken at 200 rev/min for 30 min, and filtered. The organic nitrogen content in the extracts was estimated by dry combustion using a CN analyzer (Elementar Analysensysteme GmbH, Hanua, Germany). The microbial biomass nitrogen (MBN) was calculated as: MBN = [(organic N in fumigated soil–organic N in non-fumigated soil)/K_E_], where K_E_ is 0.57 (Jenkinson, 1988) [54].

The DON content was determined using the procedures described by Gigliotti et al. (2002) [55]. Briefly, 10 g fresh soil was mixed with 50 mL water, shaken for 1 h on a reciprocal shaker, and then the colloidal suspension was centrifuged at 15,000× *g* rev/min at 25 °C for 10 min. Next, the translucent solution was passed through a 0.45 μm membrane filter. The DON in the filtrate was determined by a continuous flow CN analyzer (Elementar Analysensysteme GmbH, Hanua, Germany). The mineral N (NO_3_^−^ and NH_4_^+^) content in soil was estimated using the KCl extraction method [56]. Briefly, 10 g fresh soil was suspended with 2 M KCl solution at 1:10 soil-to-solution ratio using a reciprocal shaker at 180 rev/min for 30 min. The extract was filtered, and the concentration of NO_3_^–^ and NH_4_^+^ in the filtrates were determined using a continuous flow automated colorimeter (AA3, Automatic chemical analyzer, Easychem Plus, Europe).

### 2.5. Statistical Analysis 

Data of all soil parameters were presented on an oven-dried weight basis. Analysis of variance (ANOVA) was performed using the SPSS Statistics 25.0 software package (SPSS Inc. IBM, Chicago, IL, USA). The main treatment effects on variable means were detected and compared using the least difference (LSD) at the 0.05 probability level.

## 3. Results 

### 3.1. Soil Total N and Organic C

Five-year continuous substitution of chemical fertilizer with organic manure treatments (OM and OM + NF) significantly improved the soil total N (STN) contents in topsoil depths (Figure 1A). The average STN contents of OM, OM + NF, and NF treatments were increased by 32.9% (0.27 g kg^−1^), 8.4% (0.07 g kg^−1^), and 8.7% (0.07 g kg^−1^), respectively, when compared with the CK. The OM treatment had the significantly highest STN contents in 0–30 cm topsoil depths, which was an average 32% (0.41 g kg^−1^), 46.8% (0.54 g kg^−1^), and 50.3% (0.57 g kg^−1^) greater than OM + NF, NF, and CK treatments, respectively. The STN contents of OM + NF treatment at 0–10 and 10–20 cm topsoil depths were 19.1% (0.24 g kg^−1^) and 17.53% (0.22 g kg^−1^) greater over CK, and significantly higher than NF treatment (Figure 1A). The STN variations between NF and CK treatments were identical (*p* > 0.05) except at 40–50 cm depth of the profile.

Soil organic C (SOC) differences among treatments were significant across the profile depths, and SOC contents were higher with OM and OM + NF treatments (Figure 1B). The average SOC contents of OM, OM + NF, and NF treatments increased by 65.2% (4.42 g kg^−1^), 38% (2.57 g kg^−1^), and 21.6% (1.46 g kg^−1^), respectively, when compared to the CK. Although all treatments demonstrated higher SOC concentrations in 0–20 cm topsoil depths, the SOC contents of OM and OM + NF treatments were significantly greater (*p* < 0.01) than NF and CK treatments. The OM, OM + NF, and NF treatments had an average of 90% (9.85 g kg^−1^), 52.9% (5.78 g kg^−1^), and 15.8% (1.72 g kg^−1^) higher SOC in 0–20 cm topsoil depths. A gradual declining trend in SOC contents with increased profile depths was observed under all treatments.

### 3.2. Soil Organic C/TN Ratio

Significant variations were observed among fertilizer substitution treatments on soil organic C/TN ratio across the selected profile, and treatment effects were higher in topsoil layers (Figure 1C). The average soil organic C/TN ratio of OM, OM + NF, and NF treatments were increased by 22.4% (1.99), 24.1% (2.14), and 10.5% (0.93), respectively, when compared to the CK. The soil organic C/TN ratio of each depth under OM and OM + NF treatments were statistically similar, and organic C/TN ratio differences among OM, OM + NF, and NF treatments at 0–10, 10–20, and 20–30 cm soil depths were non-significant. Under all treatments, we found a gradual declining trend in soil organic C/TN ratio with an increase in profile depths.

### 3.3. Labile Organic N Pools

The variations among fertilizer substitution treatments on the depth distribution of soil labile organic N fractions in the profile were statistically significant (Figure 2A–C). The particulate organic N (PON) differences among treatments were pronounced in 0–20 cm topsoil layers, while a slight variation was observed at 40–50 and 50–60 cm subsoil depths (Figure 2A). The OM and OM + NF treatments had 106.5% (0.49 g kg^−1^) and 56.5% (0.26 g kg^−1^), and 93.5% (0.43 g kg^−1^) and 47.8% (0.22 g kg^−1^) higher accumulation of PON at 0–10 and 10–20 cm soil layers, respectively, when compared to the CK. The NF treatment did not significantly affect PON contents at 0–10 and 10–20 cm topsoil layers. The observed PON differences among treatments were statistically identical at 20–30, 30–40, 60–70, 70–80, 80–90, and 90–100 cm soil profile depths.

Irrespective of soil depths, the MBN differences among treatments were significant, and MBN contents were higher with OM and OM + NF treatments (Figure 2B). The MBN contents under OM and OM + NF treatments at 0–10 and 10–20 cm topsoil depths were identical but significantly highest (*p* < 0.01) over NF and CK treatments. The average MBN contents of OM, OM + NF, and NF treatments were increased by 32.8% (8.14 mg kg^−1^), 39.4% (9.77 mg kg^−1^), and 2.9% (0.73 mg kg^−1^), respectively, when compared to the CK. We found a gradual declining trend in soil MBN contents with increased in profile depths under all treatments.

The variations among fertilizer substitution treatments on soil DON contents were significant across the profile depths (Figure 2C). The DON contents in 0–30 cm topsoil depths of OM and OM + NF treatments were significantly higher (*p* < 0.01) than other treatments, while NF treatment deposited significantly highest (*p* < 0.01) DON in 40–100 cm subsoil depths (Figure 2C). The average DON contents in 0–30 cm profile depths under OM, OM + NF, and NF treatments were increased by 77.3% (40.77 mg kg^−1^), 46.9% (24.72 mg kg^−1^), and 20.1% (10.61 mg kg^−1^), respectively, when compared to the CK. In contrast, the average DON content in 40–100 cm subsoil depths of NF treatment was 53.7% (31.75 mg kg^−1^), 49.1% (29.91 mg kg^−1^), and 83.2% (41.26 mg kg^−1^) greater than OM, OM + NF, and CK treatments, respectively. The DON contents at 30–40 cm depth of OM, OM + NF, and NF treatments were identical but significantly higher (*p* < 0.01) than CK treatment. 

### 3.4. Mineral N Pools

The variations among treatments on NO_3_^−^ and NH_4_^+^ content were significant across the selected profile (Figure 3A,B). The NF treatment had the highest (23.9 mg kg^−1^) average NO_3_^−^ content in the profile, which was 21.4% (4.36 mg kg^−1^), 13.1% (3.33 mg kg^−1^), and 696.7% (20.89 mg kg^−1^) higher than OM, OM + NF, and CK treatments, respectively. However, the average treatment effect on NO_3_^−^ content in 0–30 cm topsoil layers was OM > OM + NF > NF > CK while below 30 cm soil layers NO_3_^−^ accumulation followed NF > OM + NF > OM > CK (Figure 3A). Compared to CK, the average NO_3_^–^ content in 0–30 cm topsoil layers of OM, OM + NF, and NF treatments were increased by 912.9% (33.71 mg kg^−1^), 532.1% (19.65 mg kg^−1^), and 180% (6.65 mg kg^−1^). On the other hand, the NF soil accumulated 85.8% (187.53 mg) of profile total NO_3_^−^ in 30–100 cm profile depths, which was an average 145.6% (17.63 mg kg^−1^), 47.63% (9.51 mg kg^−1^), and 998.7% (27.3 mg kg^−1^) higher than OM, OM + NF, and CK treatments, respectively. The NO_3_^−^ content differences among OM, OM + NF, and NF treatments at 30–40 cm depth were non-significant. 

Irrespective of soil depths, NH_4_^+^ contents of OM treatment were significantly highest (Figure 3B). The average NH_4_^+^ content of OM treatment was 2.52 mg kg^−1^, which was 55.9% (0.86 mg kg^−1^), 51.4% (0.84 mg kg^−1^), and 36.7% (0.67 mg kg^−1^) greater than OM + NF, NF, and CK treatments, respectively. The NH_4_^+^ content of OM + NF treatment was significant (*p* < 0.05) only at 0–10 and 10–20 cm topsoil depths (Figure 3B), an average 26.4% (0.5 mg kg^−1^) and 24.1% (0.47 mg kg^−1^) higher than NF and CK treatments, respectively. However, the average NH_4_^+^ contents OM + NF and NF treatments in 20–100 cm soil depths were 64.4% (0.96 mg kg^−1^) and 50.4% (0.82 mg kg^−1^) lower than OM treatment, respectively. Soil NH_4_^+^ variations between NF and CK treatments were identical (*p* > 0.05) in all depths of the selected profile.

### 3.5. Correlations among STN and Its Fractions 

The STN was positively and significantly correlated with its labile organic N (PON, MBN, and DON) and mineral N (NO_3_^−^ and NH_4_^+^) fractions in topsoil layers (0–10 and 10–20 cm) (Table 1). Among the labile organic N fractions, the PON showed the highest correlation with STN in the topsoil layers, followed by DON and MBN. The correlation between STN and NO_3_^−^ was more significant than between STN and NH_4_^+^. The labile organic N fractions showed a higher correlation with NO_3_^−^ than NH_4_^+^ in topsoil layers. However, correlations among the parameters were weak and non-significant in most depths below 20 cm of the selected soil profile.

## 4. Discussion 

### 4.1. Effect of Fertilizer Substitution Treatments on STN, SOC, and Organic C/TN Ratio 

Our study suggests that chemical to organic fertilizer substitutions (OM and OM + NF treatments) increases soil total N (STN) content in topsoil layers which is in agreement with previous studies [22,43,57]. The possible reason can be attributed to the significant influence of continuous manuring on soil N fractions because continuous manure inputs (exclusive or combined with chemical fertilizer) increase soil organic matter (SOM) content [22,43]. SOM could enhance soil microbial abundance and activity by providing C sources for microbial metabolism [58]. As the SOM decomposes, it releases particulate organic matter (POC and PON) and dissolved organic matter (DOC and DON) along with mineral nutrients [59,60,61]. The leftovers (concentrations) of those nitrogenous compounds (PON, MBN, and DON) that remained in the topsoil profile following soil–microorganism–plant interactions were higher under the OM and OM + NF treatments, likely contributed to STN storage. The positive and significant correlations found among STN and its fractions support this assumption. Previous research has found that the STN content is closely related to soil N availability, which is consistent with our findings [62]. However, we did not find a major improvement in STN content under the NF soil profile. Previous research found that in a chemically fertilized maize-based cropping system, aboveground N uptake was significantly greater than N immobilization [63]. Moreover, the SOC content under the NF soil profile was substantially lower compared to the OM and OM + NF soil profiles. Therefore, high aboveground N uptake by crops along with low SOC content possibly affected the microbial N immobilization of NF soil. Moreover, due to maize residue removal (in the current study), crop litter or root exudation contributions under NF treatment might be limited to maintaining SOM status and supporting soil aggregation (due to reduced PON content). As a result, most fertilizer N that remains at the topsoil profile after crop uptake and microbial utilization is perhaps not immobilized but leached down to subsoil layers or contributed to other soil N loss pathways.

Composed cattle manure is a rich source of organic C. Therefore, by substituting N fertilizer with manure-containing treatments (OM or OM + NF) in the wheat–maize growing system for five years, we found significant increase in SOC contents in the profile, specifically in 0–20 cm topsoil depths. Many previous studies with organic–inorganic fertilization also reported substantial improvement in SOC with manure-containing treatments compared to the exclusive chemical fertilization [17,23]. Our results found an improvement (*p* < 0.05) in SOC contents in 20–50 cm subsoil depths of NF treatment than CK. The possible reason might be attributed to increased rhizodeposition and belowground biomass production of wheat–maize cropping system under N fertilization. However, average SOC contents in the profile of OM and OM + NF treatments were 35.9% (2.95 g kg^−1^) and 13.5% (1.11 g kg^−1^) greater than the NF treatment. Such SOC changes explained in this paper indicate that organic manure substitutions for chemical N fertilizer can play a key role in improving soil fertility and productivity through soil C sequestration.

Soil organic C/TN ratio represents the interaction of soil C and N cycling and the stability of SOM [64]. Nitrogen fertilization increases SOM stock in maize-based cropping systems by influencing net primary production and rhizodeposition, affecting soil C/N ratio through SOC changes [65]. Thus, irrespective of N sources applied (OM, OM + NF, or NF), we found an improvement in soil organic C/TN ratios in the profile. Still, the average soil organic C/TN ratio of the OM and OM + NF treatments was higher than the NF treatment. The main reason for that is likely the constant input of high C/N organic sources, because manure-containing treatments (OM and OM + NF) increased the average SOC content in the profile substantially higher than NF treatment alone. However, translocation of high C/N organic compounds from topsoil layers to subsoil layers with percolating water from irrigation or rainfall could contribute to higher organic C/TN distribution in subsoil layers as found under OM and OM + NF treatments. Meanwhile, deposition of clay associated low C/N containing SOM clay fractions in subsoil layers likely came up with the gradual declination of soil organic C/TN ratios with an increase in profile depths as previously suggested [25,66].

### 4.2. Effect of Fertilizer Substitution Treatments on Labile Organic N Pools

After a 5-year substitution of chemical fertilizer with organic manure, we found significant (*p* < 0.01) improvement in PON content mostly at topsoil depths (0–10 and 10–20 cm) of the profile. Qiu et al. (2016) and Hai et al. (2010) also found higher PON content with manure containing treatments than chemical fertilization alone [43,67]. The PON contents were highest with OM treatment, second by OM + NF treatment, and marginal with NF treatment, indicating that the organic manure was the primary factor affecting N concentration in particulate organic matter (POM). Previous studies agree with our explanation [67,68]. Manure application increases soil organic matter (SOM) content, whose turnover releases particulate organic matter (POM; POC and PON), which can remain in topsoil layers for several years due to their short turnover time (<10 years) [22,59,69,70]. Moreover, POM being associated with clay minerals promotes soil aggregation, thus could reduce PON turnover by strengthening its physical protection against microbial oxidation [44]. Hence N concentration in aggregate associated POM is greater than free POM [67]. These findings indicate that higher PON accumulation in topsoil layers of OM and OM + NF treatments likely facilitated through manure-induced soil aggregation. However, soils under NF and CK treatments probably had reduced SOM and soil aggregates due to limited organic C sources. Consequently, lower PON contents were observed in those profiles. 

The differences in soil MBN content among fertilizer substitution treatments were significant across the selected profile. MBN contents were higher with manure-containing treatments in all sampling depths, and variations between OM and OM + NF treatments were insignificant except at 20–30 cm soil depth. Conversely, the NF treatment had no significant influence on soil MBN content over the CK treatment. A substantial increase in soil MBN contents with organic manure and organic–inorganic combined fertilizer treatments were reported by Guo et al. (2019) [57]. Liang et al. (2011) and Qiu et al. (2016) also found significantly higher MBN content in topsoil depths under combined fertilization [43,71]. Manure application increases soil microbial abundance and diversity, primarily by improving organic C availability for microbial metabolism [22,46,69]. Moreover, the highest SOC content (found in this study) and constant SOC turnover rate were resultant with organic (manure) or organic–inorganic combined fertilizer treatments [57]. These findings indicate that OM and OM + NF treatments likely supplied readily mineralizable organic carbon for microbial metabolism in a relatively consistent manner, possibly the main reason for higher MBN contents in those soils. Moreover, conventional tillage (practiced in this study) perhaps accelerated manure effects on microbial growth and activity by increasing aeration. Therefore, applying OM and OM + NF treatments, we found the highest MBN concentrations at topsoil depths (0–10 and 10–20 cm). The higher SOM decomposition, soil respiration, enzyme activities, and biomass content as recorded in topsoil layers of previous studies [72], further supports our assumption. However, some dissolved organic C (DOC) in manure soils can be deposited to deeper layers by earthworm borrow, decaying root holes, and leaching [73]. Such deposition of DOC likely leads to microbial biomass distribution in the subsoil layers. Apparently, soil microbial growth and activity were reduced by limited microbial resources (organic C, in particular) under NF or CK treatments. As a result, no significant improvement in MBN content was noticed in those profiles.

The DON contents in topsoil depths were substantially increased with OM and OM + NF treatments, while in subsoil depths (below 40 cm), DON contents were dominated (*p* < 0.01) by NF treatment. These variations among treatments clearly indicate that substituting organic manure for chemical fertilizer improves soil DON content and increases the DON retention capacity of the soil. SOM is the major source of DOM (DOC and DON) [61]. The DOM being released from decomposing SOM could incorporate into the soil aggregates due to their reactive nature to soil particles [44,60]. Therefore, improved soil aggregation and aggregate stability under manure treatment likely promoted higher accumulation of DON in topsoil layers of OM and OM + NF treatments. However, DON is highly mobile and significant DON leaching was often reported under N fertilization [34,40]. We also found significant DON deposition in all selected depths of OM, OM + NF, and NF treatments over the CK (control). However, average DON deposition in 40–100 cm subsoil depths of chemical fertilizer (NF) treatment was 49.1% (29.91 mg kg^−1^) to 53.7% (31.75 mg kg^−1^) greater than its organic substitutes. In other words, chemical fertilizer substitution with organic fertilizer substantially reduced potentially leachable DON content in the profile. Although root exudates and turnover are likely the primary sources of DON in chemically fertilized agricultural soils, the exact mechanism of how chemical fertilizer influences DON distribution is unclear [74,75]. Still, less availability of leached DON for plant and microbes than mineral N forms, as previously found in agroecosystems [40], could cause DON accumulation in subsoil layers. 

### 4.3. Effect of Fertilizer Substitution Treatments on Mineral N Pools

Our research indicated that NO_3_^−^ contents in 0–30 cm topsoil depths were increased (*p* < 0.01) with chemical to organic fertilizer substitution (OM or OM + NF), while NO_3_^–^ depositions in 40–100 cm subsoil depths were enhanced (*p* < 0.01) with exclusive nitrogen fertilizer (NF) treatment. Organic substitutions retained significantly higher NO_3_^−^ in surface layers, possibly for manure-induced enhanced soil aggregation and SOM content, which could provide some physical protection against leaching. Moreover, soil NO_3_^−^ held in the aggregates is covered from microbial reduction, which may further assist in retaining NO_3_^−^ in the topsoil profile. Conversely, the relatively poor soil conditions under continuous chemical fertilization likely had inadequate physical protection against NO_3_^−^ leaching from the persistent downward flow of water from precipitation or irrigation, because percolating water can readily translocate NO_3_^−^ to deep soil layers [76]. As a result, the significantly highest NO_3_^−^ accumulation occurred in subsoil layers of chemical fertilizer treatment. 

Although soil NH_4_^+^ has multiple fates, nitrification is believed to be a single major fate of available NH_4_^+^ due to cultivated soils’ high nitrification potential [77]. Moreover, soil microbes prefer NH_4_^+^ for N assimilation than other N forms [78]. Therefore, lower available NH_4_^+^ contents compared to NO_3_^−^ were often reported in soil fertility studies [22,43]. The report presented in this paper is also in line with previous studies. In our selected profiles, the depth distribution of available NH_4_^+^ ranged between 1.34 and 2.96 mg kg^−1^ soil. Still, OM treatment contained significantly higher NH_4_^+^ in all selected depths. Gradually released NH_4_^+^ from organic manure likely supported more active microbial biomass with greater N demand that was probably met by NO_3_^−^ immobilization. Thus, the turnover of high microbial biomass could contribute to the soil NH_4_^+^ pool. Furthermore, because NH_4_^+^ and K^+^ compete for the same ion-exchange sites of 2:1 clay minerals due to their identical size and valence properties [79,80], the potassium (K) content (8.0 g kg^−1^) of applied manure could also contribute to the release of clay fixed NH_4_^+^ in subsoil depths of OM treatment.

### 4.4. The Correlations among STN and Its Fractions

The correlations among STN and its fractions were positive and significant only at 0–10 and 10–20 cm surface layers (Table 1), most possibly for topsoil properties (the higher microbial abundance and activity with enhanced resource availability) that support SOM decomposition and nutrient mineralization [81,82]. These correlations indicate that STN was the primary determinant of profile’s labile organic N (PON, MBN, and DON) and mineral N (NO_3_^−^ and NH_4_^+^) content. In other words, changes (positive or negative) in labile organic N and mineral N pools by soil management practices could impact soil total N stocks. As microbial turnover of particulate organic matter releases dissolved organic matter, that influences soil microbial biomass [59,83,84]; the positive and significant correlations found among PON, MBN, and DON confirm that they are closely linked.

## 5. Conclusions

Our observations revealed that chemical N fertilizer, when 100% substituted with organic manure, exhibited the most significant improvements in STN, SOC, labile organic N (PON, MBN, and DON), and mineral N (NO_3_^−^ and NH_4_^+^) content of the profile, especially in 0–30 cm topsoil layers. Organic manure combined with chemical fertilizer (50% substitution) moderately improved topsoil labile organic and mineral N pools, still significant over chemical N fertilizer treatment. Application of chemical fertilizer alone showed little or no improvement in STN, PON, and MBN content of the profile but significantly increased DON and NO_3_^–^ concentration in subsoil layers leading to a potential risk of N leaching and groundwater contaminations. As labile organic pools are the early indicators of long-term changes in stabilized nutrient pools, our findings suggest that chemical fertilizer substitution with organic manure (100% or 50%) could improve the sustainability of intensively managed farming systems by improving labile organic N and mineral N pools while reducing the potential risk of N leaching. However, we recommend 50% organic substitution for chemical fertilizer because it improves topsoil N pools significantly as well as substantially reduces leachate (DON, NO_3_^−^, and NH_4_^+^) deposition in deep soil.

## Figures and Tables

**Figure 1 ijerph-18-12848-f001:**
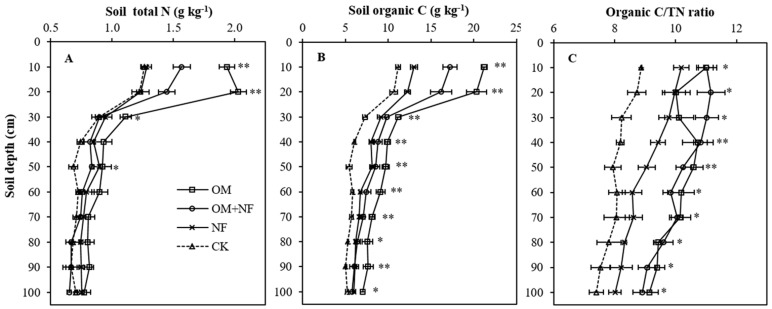
Effect of five years of continuous N equivalent fertilizer substitution treatments on the depth distribution of soil total N (**A**), soil organic C (**B**), and organic C/TN ratio (**C**) in the profile. ** Significant differences among treatments at *p* < 0.01, * significant differences among treatments at *p* < 0.05. OM, organic manure; OM + NF, organic manure with nitrogen fertilizer; NF, nitrogen fertilizer; CK, control.

**Figure 2 ijerph-18-12848-f002:**
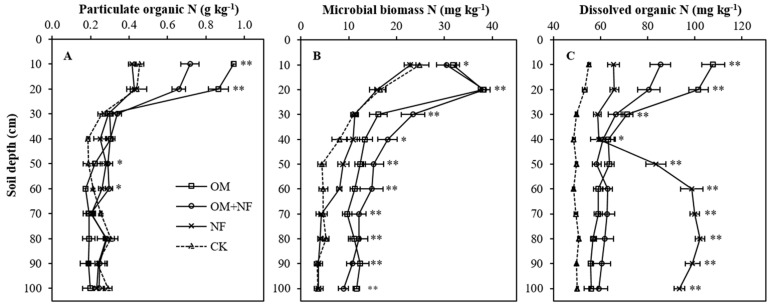
Effect of five years of continuous N equivalent fertilizer substitution treatments on the depth distribution of particulate organic N (**A**), microbial biomass N (**B**), and dissolved organic N (**C**) in the profile. ** Significant differences among treatments at *p* < 0.01, * significant differences among treatments at *p* < 0.05. OM, organic manure; OM + NF, organic manure with nitrogen fertilizer; NF, nitrogen fertilizer; CK, control.

**Figure 3 ijerph-18-12848-f003:**
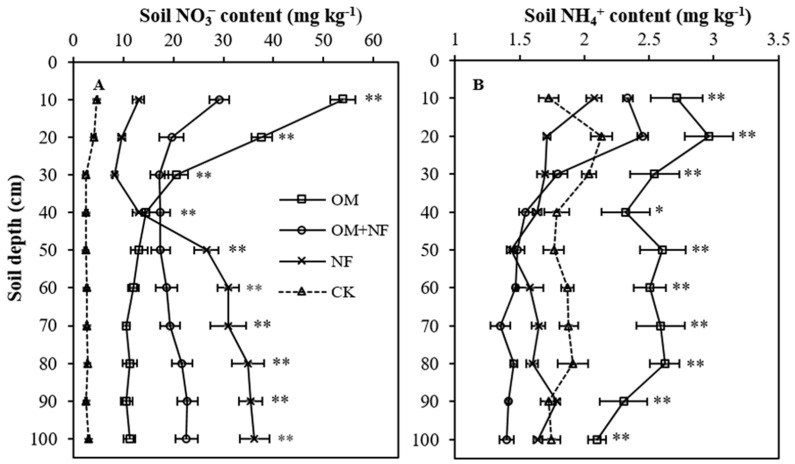
Effect of five years of continuous N equivalent fertilizer substitution treatments on the depth distribution of NO_3_^−^ (**A**) and NH_4_^+^ (**B**) in the soil profile. ** Significant differences among treatments at *p* < 0.01, * significant differences among treatments at *p* < 0.05. OM, organic manure; OM + NF, organic manure with nitrogen fertilizer; NF, nitrogen fertilizer; CK, control.

**Table 1 ijerph-18-12848-t001:** Correlations (Pearson’s) among STN and its fractions in 0–10 and 10–20 cm soil depths.

Parameter ^a^	STN	PON	MBN	DON	NO_3_^−^	NH_4_^+^
0–10 cm						
STN	1					
PON	0.952 **	1				
MBN	0.795 **	0.861 **	1			
DON	0.800 **	0.747 **	0.631 *	1		
NO_3_^–^	0.903 **	0.933 **	0.802 **	0.845 **	1	
NH_4_^+^	0.728 **	0.795 **	0.805 **	0.789 **	0.896 **	1
10–20 cm						
STN	1					
PON	0.901 **	1				
MBN	0.752 **	0.839 **	1			
DON	0.760 **	0.692 *	0.621 *	1		
NO_3_^−^	0.934 **	0.883 **	0.848 **	0.889 **	1	
NH_4_^+^	0.844 **	0.842 **	0.802 **	0.561	0.787 **	1

^a^ STN, soil total N; PON, particulate organic N; MBN, microbial biomass N; DON, dissolved organic N. ** Significant at *p* < 0.01, * Significant at *p* < 0.05.

## Data Availability

The data presented in this study are available on request from the corresponding author. The data are not publicly available because they belong to an ongoing project which has not finished yet.

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
