# Peer review of "Substitution of Chemical Fertilizer with Organic Fertilizer Affects Soil Total Nitrogen and Its Fractions in Northern China"

_ijerph, 2021, doi:10.3390/ijerph182312848_

Round 1

Reviewer 1 Report

It was a pleasure to read and inquire about the situation of fertilizers in the territories of northern China and the possibility of replacing chemical fertilizers with organic ones in the perspective of sustainable agriculture.

The work was done well and with methodological rigor.
Just a few tips:

  • the introduction should be scaled down and should briefly place the study in a broad context. The paragraph is very broad, it could be reduced.
  • -If the materials and methods have been used in other studies it is good to use the bibliographic reference.
    line 158 FAO-Unesco, insert bibliographic reference
    both the results and the discussions are quite good and argued with recent literature.
  • The bibliographic references in the text must absolutely be reviewed, they must be inserted by numbers in order of appearance and placed between [  ]

Reviewer 2 Report

The paper evaluated the effects of organic and chemical fertilizers on soil nitrogen in agricultural region in northern China. The results in the paper are presented clearly and discussion is sufficient. It will provide valuable information for revealing soil N management of intensive farming systems. It can be accepted after a minor revision.

Authors talked changes on soil C/N in the paper, but no information about soil organic carbon (SOC), so they need to add SOC data to the results, the discussion about changes in SOC also need to be made.

Reviewer 3 Report

The research topic is largely researched. In the introduction, there are many unnecessary generalities. It needs to be rewritten. The methodology did not state whether the crops were harvested each year. Very very old analytical methods. There are no standards at all. Other units in the figure and the description only in%, it is not correct. The work was written in a very bad style. The sentence is very long and confused. Too many citations in each sentence. The cited names may be included at the end of the assignment. Don't break the sentence into pieces. As a result, the content of the sentences is hardly understandable. I have marked a lot of comments in the text. References written not in accordance with the requirements of the editorial office. The work requires thorough editing. 

Reviewer 4 Report

This is the revision of the manuscript number ijerph-1412248 Title: “Substitution of chemical fertilizer with organic fertilizer affects 2 soil total nitrogen and its fractions in northern China”, proposed by Miss Bianca Raluca Berezoschi and colleagues for consideration for publication in International Journal of Environmental Research and Public Health.

The manuscript raises a novel theme and attempts to address an increasingly problematic issue, the contamination of underground layers and the loss of quality of food-producing soils, the ideas proposed on the application of organic amendments to address the mineral N content they seem excellent in the manuscript. However, some suggestions to improve the manuscript should be considered.

Material and methods:

It would be important to indicate the characteristics of composted cattle manure,

Line.165. ¿270 kg ha-1 N ? It would be convenient to apply the amount of N based on the extractions of the crops to be exploited, corn and wheat, reviewing the diverse existing bibliography, not based on the conventional use of farmers in the area, which could not have any scientific rigor.

L.190. Due to the high mobility of N as expressed in the introduction. Why is an annual sampling not carried out, after the end of each crop cycle?

L.109. Once the author's quote is placed, I suggest not to immediately show the same author's quote again without commenting on anything different about that quote. Example: “described by Abd El-Fattah (Abd El-Fattah et al., 2013)”, Correct: “described by Abd El-Fattah et al. (2013)”

The crop residues after each cycle could influence the properties of the soil, it should be indicated that it is done with the crop residues once the harvest is done.

  1. 202. As a recommendation, there is a letter change in the authorship of the Kjeldahl method.

Results and discussion:

L.334. Table 1. As a recommendation, it would be nice to place the depth data of the samples shown in the characterization in the same position. (10-20 cm)

In the study you analyze the microbial mass, this study is very extensive in terms of the repercussion towards the nitrification and denitrification of N, I suggest that you carry out a study of the enzyme nitrogenase activity, since its concentration has a high influence on the process of N on the ground

General comments:

The author reports knowledge about the use of organic amendments to supplement the application of mineral fertilizer in the soil, I wonder if the application of composted cattle manure affects the quality and quantity of the agronomic performance of corn and wheat, which could be parameters of great interest , because these cereals are the most consumed by the population worldwide

Round 2

Reviewer 3 Report

Some the comments were corrected, some were not. The lack of amendments has not been clarified. There are new bugs in the added parts. Even badly writable units. the comments marked in the text, 18. I leave the final decision regarding the acceptance of the work for publication to the editor-in-chief. 

The old summary has been left in the review form.
